# Transcription-driven DNA supercoiling counteracts H-NS-mediated gene silencing in bacterial chromatin

Nara Figueroa-Bossi [1], Rocío Fernández-Fernández [2], Patricia Kerboriou[1], Philippe Bouloc [1], Josep Casadesús [3,4], María Antonia Sánchez-Romero [2] & Lionello Bossi [1] ✉

In all living cells, genomic DNA is compacted through interactions with dedicated proteins and/or the formation of plectonemic coils. In bacteria, DNA compaction is achieved dynamically, coordinated with dense and constantly changing transcriptional activity. H-NS, a major bacterial nucleoid structuring protein, is of special interest due to its interplay with RNA polymerase. H-NS:DNA nucleoprotein filaments inhibit transcription initiation by RNA polymerase. However, the discovery that genes silenced by H-NS can be activated by transcription originating from neighboring regions has suggested that elongating RNA polymerases can disassemble H-NS:DNA filaments. In this study, we present evidence that transcription-induced counter-silencing does not require transcription to reach the silenced gene; rather, it exerts its effect at a distance. Counter-silencing is suppressed by introducing a DNA gyrase binding site within the intervening segment, suggesting that the long-range effect results from transcription-driven positive DNA supercoils diffusing toward the silenced gene. We propose a model wherein H-NS:DNA complexes form in vivo on negatively supercoiled DNA, with H-NS bridging the two arms of the plectoneme. Rotational diffusion of positive supercoils generated by neighboring transcription will cause the H-NS-bound negatively-supercoiled plectoneme to "unroll" disrupting the H-NS bridges and releasing H-NS.

The structure and biological activity of genomic DNA, in all forms of life, are subordinate to the need for fitting into a space considerably smaller than the volume that the DNA would occupy if unconstrained. Even in prokaryotes, where chromosomal DNA is not surrounded by a nuclear envelope, it occupies only a portion of the cell volume, termed the nucleoid[1]. DNA compaction inside the nucleoid results from various mechanisms including the action of DNA structuring proteins, DNA supercoiling by topoisomerases and pressure from molecular crowding[1,2]. Among the best-known nucleoid-associated proteins (NAPs), H-NS is a small, abundant protein found in numerous species of gram-negative bacteria[3–5]. H-NS oligomerises along the DNA, establishing contacts with the minor groove[6]. The process of oligomerisation, in the form of multimerisation of dimers, begins with binding to AT-rich, high-affinity nucleation sites[7,8]. Due to its cooperative nature, oligomerisation can extend into regions of lower affinity. The presence of two DNA-binding domains on each dimer unit enables the protein filament to bridge separate segments of a DNA molecule, a property regarded as pivotal in the H-NS ability to condense DNA[9–11].

[1]Université Paris-Saclay, CEA, CNRS, Institut de Biologie Intégrative de la Cellule (I2BC), Gif-sur-Yvette, France. [2]Departamento de Microbiología y Parasitología, Facultad de Farmacia, Universidad de Sevilla, Sevilla, Spain. [3]Departamento de Genética, Facultad de Biología, Universidad de Sevilla, Sevilla, Spain. [4]Deceased: Josep Casadesús. ✉e-mail: lionello.bossi@i2bc.paris-saclay.fr

The effect of H-NS binding on transcription is intimately linked to the nucleoid structuring function of H-NS. The H-NS:DNA filaments obstruct the access of RNA polymerase to promoters, silencing gene expression[12–16]. Due to H-NS affinity for AT-rich DNA, silencing is largely directed toward genes with a high AT content. In enteric bacteria, whose core genomes exhibit an even base composition, AT richness is the hallmark of genes acquired through horizontal transfer[17]. This led to the idea that H-NS played a central role in the evolutionary history of these bacterial species in preventing the potentially harmful effects arising from the expression of foreign DNA[18–20]. Over time, mechanisms evolved to allow the bacterium to harness the functions of foreign gene products while regulating their synthesis appropriately. Indeed, the vast majority of H-NS-silenced genes are expressed only under specific conditions thanks to the activity of regulatory proteins capable of displacing H-NS from promoter regions in response to environmental cues[21–23]. How H-NS dislodgment from a promoter leads to disassembly of the entire nucleoprotein filament in the coding sequence remains incompletely understood.

Another characteristic of AT-rich genes, particularly the high AT bias in promoter consensus sequences, is their tendency to exhibit spurious promoter activity[24]. Although typically weak, spurious intragenic promoters can sequester a significant fraction of RNA polymerase molecules if not repressed[25,26]. It is now evident that an important function of H-NS is to repress not only regular promoters at the 5′ end of genes but also the spurious promoters within coding sequences[24]. However, this extended repression, made possible by the formation of H-NS:DNA filaments, is not foolproof. Silencing pervasive intragenic transcription requires the involvement of a secondary cellular function: the highly conserved transcription termination factor Rho. In both *Escherichia coli* and *Salmonella enterica*, disrupting Rho activity via mutations or treatments leads to a marked increase in transcriptional noise within H-NS-silenced genes[27–29]. These findings imply that H-NS-bound DNA, possibly in regions of lower affinity, remains partially accessible to RNA polymerase, and that Rho plays a crucial role in halting transcription originating from spurious promoters. Rho appears to accomplish this task without specific RNA binding requirements but depending entirely on the interaction with its cofactor NusG[27]. Conceivably, the presence of the H-NS:DNA filament on the RNA polymerase path renders the elongation complex especially susceptible to Rho/NusG-mediated termination. This would be consistent with a termination pathway in which the interaction between Rho and NusG precedes and drives the interaction of Rho with the RNA[30].

In *Salmonella*, increased spurious transcription when Rho is impaired leads to the relief of H-NS silencing of pathogenicity islands[27]. This occurs due to occasional antisense transcripts originating in the region upstream of the gene encoding the *Salmonella* Pathogenicity Island 1 (SPI-1) master regulator, HilD[31]. Elongation of these transcripts toward the *hilD* promoter somehow disrupts the H-NS complex that keeps the promoter silenced, allowing the production of some HilD protein. In turn, HilD activates transcription of its own gene (by outcompeting H-NS for binding to the promoter), triggering an autocatalytic positive feedback loop that leads to further HilD accumulation and the transcription of several HilD-activated regulatory loci in SPI-1 and other islands. Remarkably, this occurs in only a fraction of the cell population presumably due the stochastic nature of the initiating event[31].

The present study investigates the mechanism by which transcription elongation interferes with H-NS silencing. In SPI-1, the genes on the 5′ side of *hilD* are all oriented opposite to *hilD*. Spurious antisense transcripts originating within these genes are predicted to be untranslated, which accounts for their susceptibility to Rho termination. We noticed that placing a gene encoding a translatable mRNA several hundred base-pairs upstream of *hilD* (and oriented toward the latter) activated *hilD* expression in cells with normal Rho activity. This response was not significantly affected by inserting a strong Rho-independent terminator immediately downstream from the inserted gene, suggesting that transcription-translation of this gene relieved H-NS silencing at a distance. Characterization of this phenomenon enabled us to link H-NS counter-silencing to the generation of DNA supercoils during transcription-translation of the upstream sequence. Here we present evidence suggesting that transcription-induced positive DNA supercoiling is responsible for destabilizing the H-NS:DNA complex at the *hilD* promoter and propose a model explaining how this could occur.

## Results

### Transcription relieves H-NS-mediated *hilD* silencing at a distance

SPI-1 genes directly activated by HilD exhibit a typical bimodal pattern of expression, characterized by the concomitant presence of ON and OFF subpopulations of bacteria[32–35]. The pattern reflects HilD's ability to activate its own expression by displacing H-NS and to maintain the activated state through a positive feedback loop[31,36,37]. In this study, HilD-mediated regulation was analyzed at the single-cell level by monitoring expression of a translational superfolder GFP (GFP$^{SF}$) fusion to the *hilA* gene, directly activated by HilD. The analysis was conducted in the background of a 28 Kb deletion that removes a significant portion of SPI-1 material on the 3′ side of *hilA* (Supplementary Fig. 1). Consistent with the results of a previous study, in strains carrying a *tetR*-P$^{tet}$ cassette positioned at various distances upstream from the *hilD* promoter, activation of the TetR-regulated P$^{tet}$ promoter with Anhydrotetracycline (AHTc) had no effect on the proportion of *hilA*$^{OFF}$ and *hilA*$^{ON}$ cells when Rho activity is unperturbed (Fig. 1a; Supplementary Fig. 2a). We showed previously that this is so because Rho-dependent termination prevents P$^{tet}$-promoted transcripts from reaching *hilD*[31]. However, a surprising picture emerged when we analysed constructs comprising the P$^{tet}$ promoter in its natural configuration fused to the *tetA* gene. In this case, activation of *tetA* transcription from positions between 600 bp and 1.6 Kb upstream of *hilD*, led to *hilA* being turned ON in the majority of cells (Fig. 1b; Supplementary Fig. 2b). These results were puzzling. The chromosomal sequences found at the insert boundaries in constructs carrying or lacking *tetA* are identical, that is, the sequence fused to P$^{tet}$ in a construct lacking *tetA* is adjacent to the 3′ side of *tetA* in the corresponding construct that contains this gene (refer to diagrams in Fig. 1a, b, and Supplementary Fig. 2a, b). Thus, in principle, there should be no reason why Rho would terminate transcription in the first set of constructs and not in the second. To settle the issue, we inserted a strong Rho-independent terminator (*hisL* Ter) at the *tetA-prgH* boundary in the strain carrying the P$^{tet}$*tetA* cassette 1.2 Kb upstream of *hilD*. Addition of the terminator results in a four-order-of-magnitude reduction in transcript levels immediately downstream of *tetA*, compared to an approximate 10-fold decrease observed when *hisL* Ter is absent (Supplementary Fig. 3). Remarkably, however, the terminator does not affect the proportion of cells in which the *hilA*-GFP$^{SF}$ fusion becomes expressed in response to P$^{tet}$ activation; this fraction corresponds to approximately 90% of cells in both cases (compare Fig. 1b, c; see Fig. 1e for compiled data from four independent experiments). Conversely, preventing translation of the *tetA* mRNA by introducing an initiator codon mutation (AUG to AAA) nearly abolishes all AHTc-induced changes (Fig. 1d). This trend is consistent with results measuring *hilD* mRNA levels in the entire bacterial population. We observed that *hilD* mRNA increases approximately 100-fold in response to AHTc treatment, regardless of the presence of the terminator, while it remains near basal levels in the strain with the *tetA* initiator codon mutation (Fig. 1f). Taken together, these results suggest that *tetA* transcription does not need to elongate all the way to *hilD* to relieve H-NS silencing; rather, it accomplishes this at a distance provided that the *tetA* mRNA is translated.

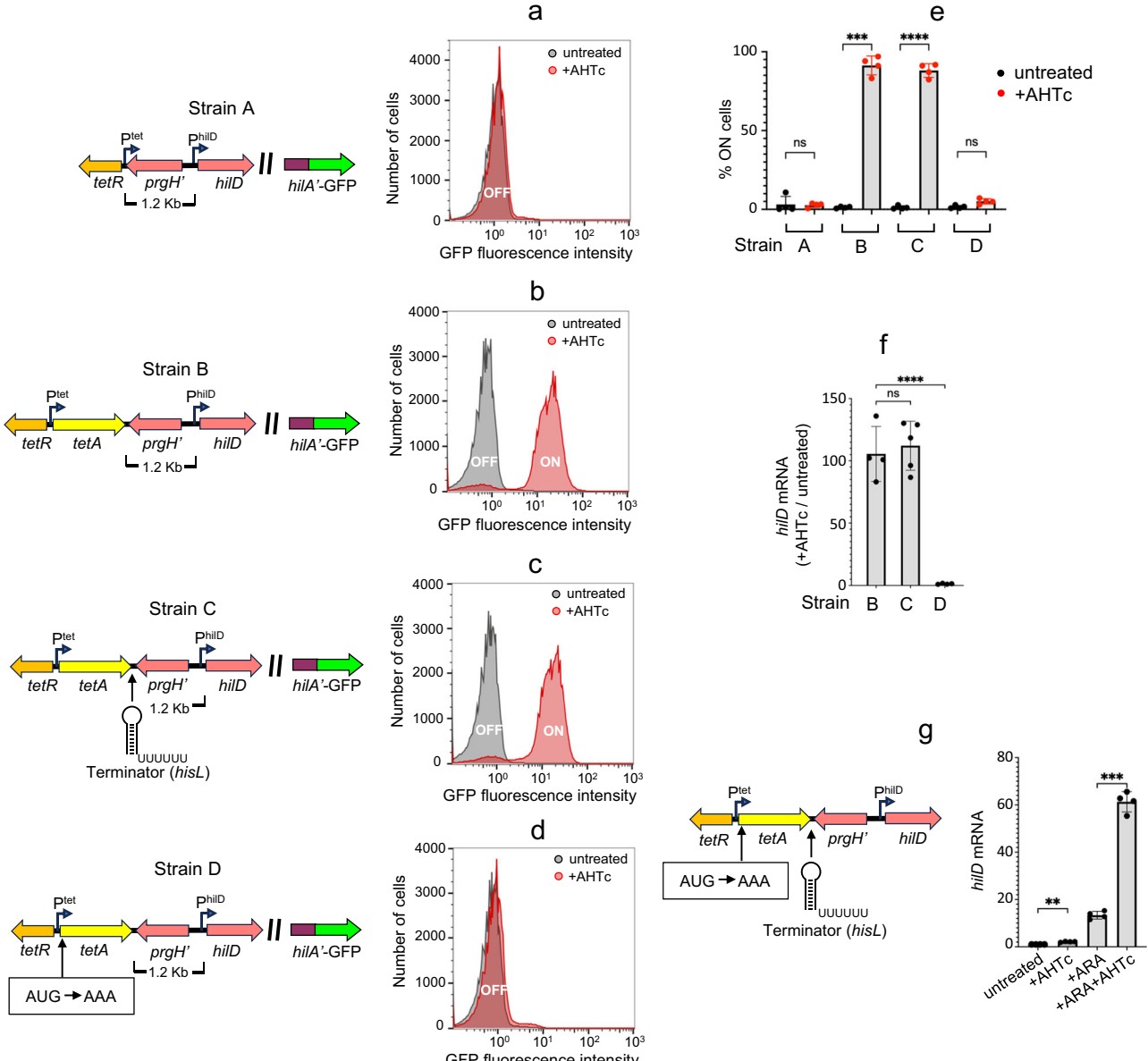

**Fig. 1 | Impact of neighboring transcription on HilD-mediated regulation.**
Strains MA14341 (strain A), MA14694 (strain B), MA14692 (strain C) and MA14403 (strain D) carry a *tetR*-P^tet cassette (strain A) or a *tetR*-P^tet*tetA* cassette (strains B, C, D) replacing the left portion of SPI-1 up to a position 1.2 Kb from *hilD* (Δ[*sitA*-*prgH*1.2]::*tetRA*) and a translational *hilA*-GFP^SF fusion replacing most of SPI-1 material to the right of *hilA* (Δ[*hilA*-STM2906]::GFP^SF; diagram in Supplementary Fig. 1). In strain C, the intrinsic transcription terminator from the histidine operon attenuator region (*hisL*) is inserted immediately downstream of *tetA*. In strain D, the translation initiation codon of *tetA* is changed to AAA. Only relevant promoters are shown (bent arrows). **a**−**d** Representative flow cytometry profiles of strains A, B, C and D, respectively, grown in the absence or in the presence of 0.4 μg/ml of AHTc. **e** Percentage of cells expressing *hilA*-GFP^SF in $n = 4$ independent repeats of the flow cytometry analysis. The error bars represent mean values ± SD. Statistical significance was determined by one-way ANOVA with Šidák's multiple comparisons test. Adjusted *P* values for the untreated/+AHTc comparison were 0.9996, 0.0004, <0.0001 and 0.2582 in strains A, B, C and D, respectively. **f** Quantification of *hilD* mRNA from cultures of strains B, C and D grown in the presence or absence of

AHTc. **g** Quantification of *hilD* mRNA from cultures of strain MA14888 grown in LB (untreated) or in LB supplemented with either AHTc, ARA or AHTc and ARA combined. Strain MA14888 carries the *tetA* initiator codon mutation in combination with the *hisL* terminator in the background of a *nusG* gene fusion to a promoter under the control of an ARA-inducible phage repressor (see text). In (**f**, **g**) RNA from $n = 4$ or $n = 5$ independent experiments was quantified by two-step reverse transcription-quantitative PCR (RT-qPCR). The RT step was carried out with a mixture of two gene-specific primers, AI69 annealing to *hilD* mRNA and AJ33 annealing to *ompA* mRNA. The resulting cDNA was amplified by qPCR with primers AI62-AI63 (*hilD*) and AJ32-AJ37 (*ompA*). Ct values were normalized to the Ct values determined for *ompA*. Error bars represent mean ± SD. Statistical significance was determined by one-way ANOVA with Šidák's multiple comparisons test. Adjusted *P* values were 0.8268 and <0.0001 for the B/C and C/D comparisons, respectively, in (**f**), and 0.028 and 0.0006 for the untreated/+AHTc and +ARA/+ARA+AHTc comparisons, respectively, in (**g**) (ns, *P* > 0.05; **$P \le 0.01$; ***$P \le 0.001$; ****$P \le 0.0001$). Source data are provided as a Source Data file.

In the absence of translation, *tetA* transcription is expected to be prematurely terminated by Rho. To clarify if the role of translation is simply to prevent Rho termination, we aimed to analyse the *hilD* response to P^tet induction under conditions where Rho is inhibited.

One such condition results from feeding arabinose (ARA) to a strain carrying the *nusG* gene under the control of an ARA-inducible repressor. ARA treatment causes this strain to become depleted of NusG, which, in turn, impairs Rho activity throughout SPI-1[27]. Note that

Rho inhibition per se already activates *hilD* expression due to runaway transcription originating from spurious promoters scattered in the regions neighboring the *hilD* gene[31]. To determine if transcription of untranslated *tetA* added to these effects, and did so at a distance, we first combined the *tetA* initiator codon mutation with the *hisL* terminator, then moved the construct into the NusG-depletable background and measured *hilD* mRNA levels in cells treated with either AHTc, ARA, or a combination of these two products. Results confirmed that AHTc alone had only a modest two-fold effect, whereas ARA led to an approximately 13-fold increase in *hilD* mRNA levels. However, in cells treated with ARA and AHTc combined, the increase was more than 60-fold (Fig. 1g). The 4.5-fold difference between the ARA-treated samples with or without AHTc is thus entirely attributable to untranslated transcripts that initiate at P[tet] and elongate up to the *hisL* terminator. This is much less than the 100-fold effect observed when *tetA* is translated (Fig. 1c), but nonetheless significant. We conclude that translation strongly contributes to the long-range effects but is not absolutely required for these effects.

To obtain direct evidence for *tetA*-mediated activation of the *hilD* promoter, we performed 5′ rapid amplification of complementary DNA ends sequencing (5′ RACE-Seq). In the particular protocol used, based on template-switching reverse transcription (TS-RT), RNA 5′ ends are defined by the non-templated addition of repeated C residues by the RT enzyme at the end of first-strand cDNA synthesis[38]. Subsequent PCR, performed semi-quantitatively, enables comparative assessments of relative levels of specific RNA species within a pool[31]. In the course of this experiment, we also measured the activity of the divergent *prgH* promoter. Results of this analysis (Supplementary Fig. 4a, b) demonstrate that exposure to AHTc triggers a substantial increase in RNAs whose 5′ ends coincide with *hilD* and *prgH* TSS. In the case of *hilD*, the magnitude of this response (Supplementary Fig. 4c) closely mirrors those measured by RT-qPCR (Fig. 1f). Note that the *prgH* promoter is thought to be activated by HilA[39]. To explain the finding that *prgH* is strongly induced as a result AHTc treatment in our *hilA*-GFP fusion background, we previously suggested that the HilA-GFP hybrid protein might retain regulatory activity[31]. However, we have since measured the effect of *tetA* transcription in a strain in which the *hilA* gene is completely deleted. We found that the *prgH* promoter is still strongly

induced in the absence of HilA (Supplementary Fig. 4d). Thus, it appears that when *prgH* activation is triggered by upstream transcription, this activation, while completely HilD-dependent[31], occurs independently of HilA.

## Counter-silencing by *tetA* transcription correlates with reduced H-NS binding across SPI-1

The impact of *tetA* transcription on H-NS binding to *hilD* and other sections of SPI-1 was evaluated through chromatin immunoprecipitation sequencing (ChIP-Seq). The strain used for this analysis (MA14443) carries the P[tet]*tetA* cassette with the *hisL* terminator (P[tet]*tetA*-T[hisL]) 1.2 Kb upstream of *hilD* and the entire SPI-1 segment to the right of the cassette (see diagram in Fig. 2a). Results from three independent experiments consistently showed a substantial reduction in H-NS binding across SPI-1 as a consequence of *tetA* transcription (Fig. 2a, b). The decrease is observed up to, but not including, the *pigAB-pphB* gene cluster. This portion of the SPI-1 is thought to have been acquired separately from the rest of the island[40,41] and is probably regulated independently. Likewise, two chromosomal loci known to bind H-NS, *proV*[42,43] and *leuO*[44], are not affected by the decrease. Conversely, two virulence loci external to SPI-1, the *pibB-virk-mig-14* cluster[45] and the *sopB* gene[46] showed reduced H-NS binding in AHTc-treated cells (Fig. 2b). We infer that these two loci are indirectly regulated by HilD. Finally, the data in Fig. 2a, b reveal substantial H-NS binding to the *tetRA* cassette under uninduced conditions. This is consistent with the cassette's relatively high AT-content (60%, *tetR*; 57%, *tetA*). Finding that the *tetA* gene responds to AHTc induction leads us to conclude that the *tetRA* DNA, although bound by H-NS, remains accessible to both the TetR protein and RNA polymerase.

## Transcription-induced counter-silencing is not specific to SPI-1

Based on the results described above, we predicted that placing the P[tet]*tetA*-T[hisL] cassette near any H-NS-silencing gene should similarly result in AHTc-dependent counter-silencing. We tested this prediction with *leuO*, a gene under strong repression by H-NS[44,47]. The P[tet]*tetA*-T[hisL] cassette was moved to a position 427 bp upstream of the *leuO* TSS (Fig. 3a). To avoid possible interference from the divergent *leuLABCD* promoter, the insertion was made by concomitantly removing a 112 bp

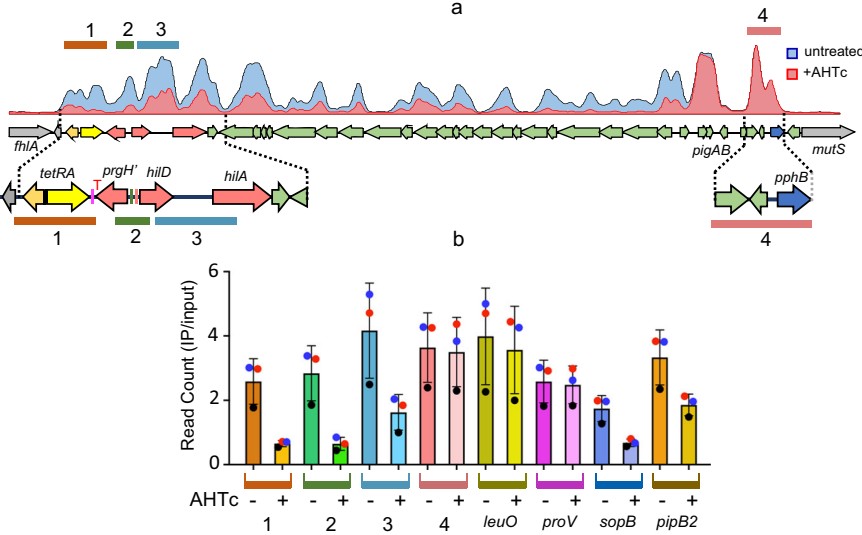

**Fig. 2 | ChIP-Seq analysis of H-NS binding to *Salmonella* chromosomal DNA.**
**a** Representative ChIP-Seq profiles from strain MA14443 (Δ[*sitA-prgH*1.2]::*tetRA*-T[hisL] *hns*-3xFLAG) grown to early stationary phase in the absence or in the presence of 0.4 μg/ml AHTc. **b** Read count quantification in selected genome sections. Read counts within the sections highlighted and numbered in (**a**) and in the H-NS-bound regions of the indicated chromosomal loci were determined with the "Samtools'

view" tool and normalized to the total reads from the entire genome. The histogram bars represent the ratios between the normalized values from IP samples and those from input DNA. The data represent the means from three independent ChIP-Seq experiments (each identified by the round symbol color, with error bars indicating standard deviations). Source data are provided as a Source Data file.

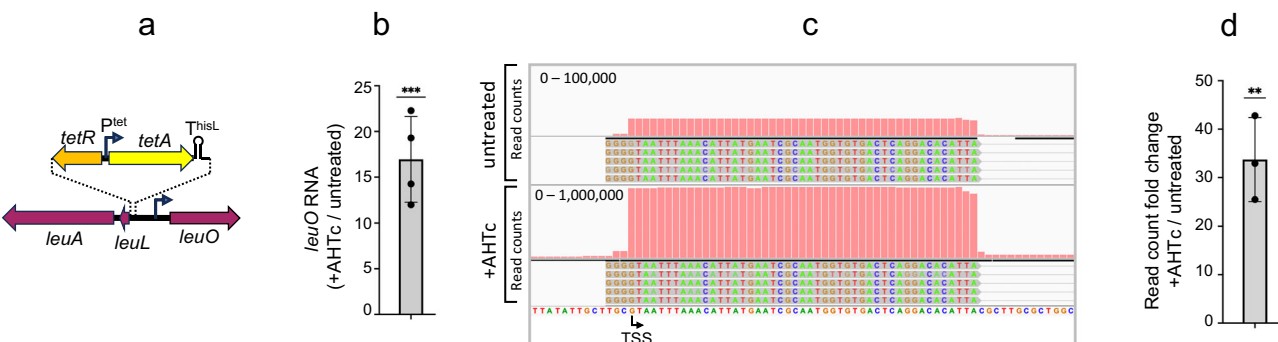

**Fig. 3 | Impact of upstream transcription on the activity of the *leuO* promoter.**
**a** Diagram showing the relevant features of strain MA14606, used in this experiment. The strain contains the *tetRA*-T$^{hisL}$ cassette replacing the promoter region of the *leuL* locus (encoding the *leu* operon leader peptide). The insertion positions the 3' boundary of the cassette 427 bp upstream of *leuO* TSS. Only relevant promoters are shown (bent arrows). **b** Quantification of *leuO* mRNA. Total RNA extracted from cultures of strain MA14606 grown in the presence or absence of AHTc (0.4 μM) was used for RT-qPCR analysis. The RT step was conducted using a mixture of primers AM99 (*leuO*) and AJ33 (*ompA*). Amplification was performed with primers AN20 and AN21 for *leuO* and primers AJ32-AJ37 for *ompA*. *leuO* Ct values were normalized to the Ct values determined for *ompA* mRNA. **c** Representative IGV snapshot from 5'RACE-Seq analysis of *leuO* mRNA. RNA was reverse-transcribed with a mixture of primers AM99 (*leuO*) and AI69 (*hilD*) in the presence of template-switching oligonucleotide (TSO) AI39. The resulting cDNA was subjected to semi-quantitative PCR amplification and high-throughput sequencing as described in Methods. Sequence reads were trimmed to remove the TSO sequence up to the terminal 3 Gs. **d** Semi-quantitative assessment of the abundance of *leuO* mRNA 5' ends. The counts of reads with the TSO sequence fused to the 5' end of *leuO* mRNA were normalized to the reads with the TSO fused to the 5' end of *hilD*. Results in (**b**, **d**) originate from $n = 4$ and $n = 3$ independent experiments, respectively. The error bars represent mean ± SD. Significance was determined by two-tailed ratio-paired *t* test. *P* values were 0,0003 in (**b**) and 0.0018 in (**d**) (**$P \leq 0.01$; ***$P \leq 0.001$). Source data are provided as a Source Data file.

fragment encompassing this promoter. The impact of *tetA* activation on *leuO* transcription was analyzed measuring *leuO* mRNA levels by RT-qPCR (Fig. 3b) and confirmed by quantifying the 5' ends of this mRNA by the 5' RACE mapping (Fig. 3c, d). These analyses show that *tetA* transcription triggers an increase of more than 10-fold in *leuO* transcription, which primarily results from activation of the *leuO* promoter. The fact that this increase is smaller than that observed at the *hilD* promoter can be explained by the absence of the auto-activation component in *leuO*, which likely enhances the *hilD* response. Possibly, the activation of *leuO* solely reflects the increased accessibility of the *leuO* promoter to the RNA polymerase holoenzyme. Overall, these findings support the idea that the transcription-dependent relief of H-NS silencing is not a specific feature of SPI-1, but rather, is a widespread phenomenon.

## Recruiting DNA gyrase suppresses transcription-induced counter-silencing

The most likely scenario to explain the long-range effect of transcription on H-NS silencing calls for the participation of DNA supercoiling. Transcription is known to induce local changes in DNA supercoiling[48–50]. During transcription elongation, frictional resistance opposes the rotation of the elongation complex around the DNA axis, which forces the DNA axis itself to rotate. This causes overwinding of the DNA helix (i.e., positive supercoiling) ahead of the moving polymerase and underwinding of the helix (i.e., negative supercoiling) behind[51,52]. In bacteria, the frictional drag resulting from the presence of ribosomes on the nascent RNA accentuates the partitioning of supercoiling domains[53,54]. The positive supercoils that accumulate in front of the RNA polymerases are typically relaxed by DNA gyrase[51]. When gyrase binding sites are absent, positive supercoils can diffuse considerable distances along the DNA and affect gene expression in neighboring regions[55,56]. We hypothesized that positive supercoils generated by *tetA* transcription may be responsible for destabilizing the H-NS:DNA complex at the *hilD* promoter. If so, placing a gyrase binding site between *tetA* and *hilD* would hinder the transmission of these supercoils to *hilD*, thereby restraining counter-silencing effects. To test this hypothesis, we inserted a 189 bp fragment encompassing the strong gyrase site (SGS) from bacteriophage Mu[57] approximately midway between *tetA* and *hilD* (433 bp and 578 bp from T$^{hisL}$ and P$^{hilD}$,

respectively, conserving the distance between *tetA* and *hilD*; Supplementary Fig. 5a, b). Introduction of the SGS led to a significant reduction in the proportion of cells that activate *hilA* upon *tetA* activation (Fig. 4a, left and center and Fig. 4b). As expected, this reduction correlated with lower accumulation of total *hilD* mRNA (Fig. 4c).

The partitioning of supercoiling domains is believed to be amplified during *tetA* transcription due to the co-translational anchoring of the emerging TetA polypeptide to the cell membrane[53]. This prompted us to test the impact of substituting the *tetA* open-reading frame (ORF) with that of the *cat* gene where the accumulation of supercoils was expected to be less pronounced, as the *cat* gene product is a cytoplasmic protein. In addition, the smaller size of the *cat* gene compared to *tetA* (657 vs 1203 bp) might further reduce the drag responsible for supercoil accumulation. Activation of *cat* transcription led to a lower proportion of *hilA*$^{ON}$ cells compared to that observed with *tetA* (Fig. 4d, left, and Fig. 4e). Notably, within the P$^{tet}$*cat* context, insertion of the SGS suppressed the *hilA* OFF-ON switching across nearly the entirety of the cell population (Fig. 4d, center, and Fig. 4e). All these changes closely correlate with the effects of AHTc exposure on *hilD* mRNA levels (Fig. 4f). Overall, these findings suggest that transcription-induced positive supercoiling directly contributes to the destabilization of H-NS:DNA complexes and counter-silencing.

## Mutations in the SGS restore counter-silencing

To confirm that the changes associated with the SGS reflected gyrase recruitment, we performed random mutagenesis of a 12 bp sequence encompassing the gyrase cleavage site within the SGS[57,58]. By screening colonies arising from this mutagenic procedure for green fluorescence emission on AHTc-containing plates, we identified clones that displayed higher fluorescence levels (Supplementary Fig. 5b). Upon sequencing, these clones were found to exhibit similar alterations in the gyrase cleavage sequence, characterized by the occurrence of five or more repeated C:G base pairs (Supplementary Fig. 5c). Analyzing two representative mutants revealed that the alterations in the SGS reversed the effects of the SGS insertion on the cell distribution, producing a significant increase in the proportion of *hilA*$^{ON}$ cells. This reversal was observed with both the P$^{tet}$*tetA* construct (Fig. 4a, right, and Fig. 4b) and the P$^{tet}$*cat* construct (Fig. 4d, right, and Fig. 4e).

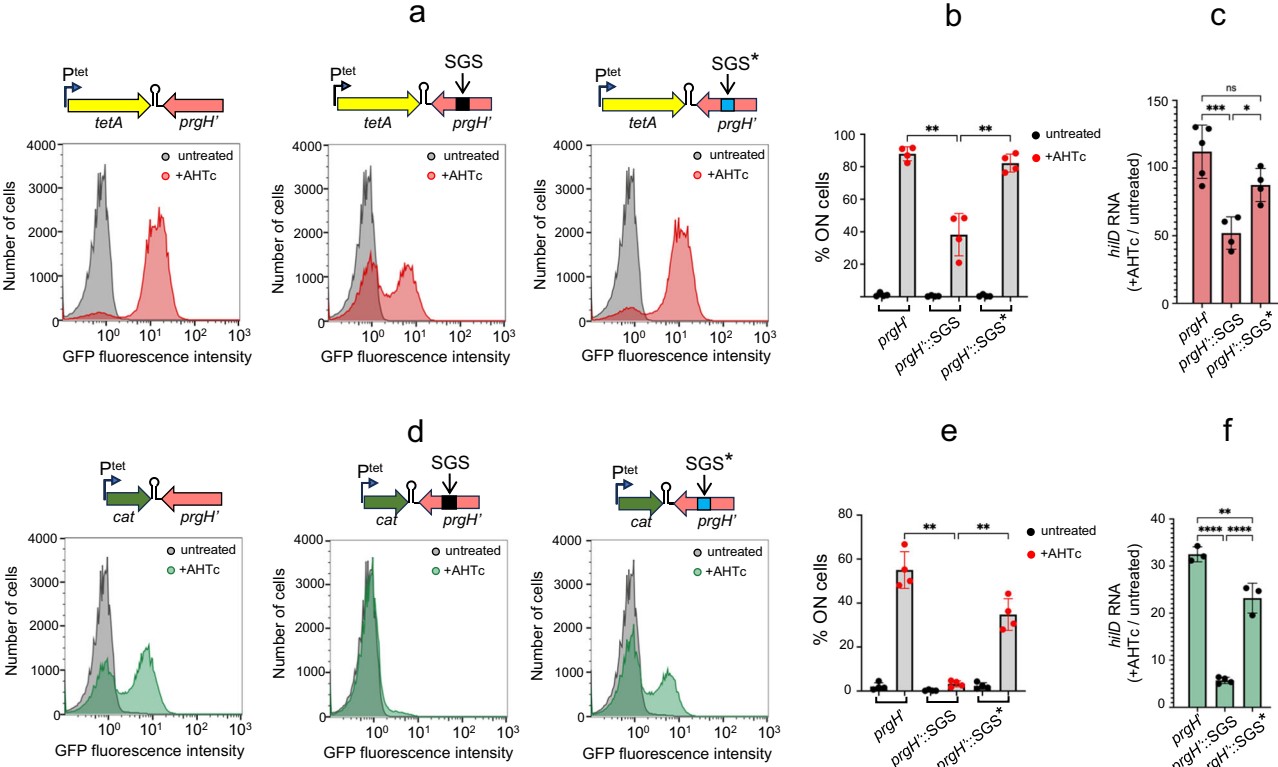

**Fig. 4 | Impact of the strong gyrase site (SGS) on transcription-induced counter-silencing. a** Representative flow cytometry profiles from cultures of strains MA14692, MA14752 and MA14793 (left, middle and right panel, respectively) grown to early stationary phase in the presence or absence of AHTc. MA14692 contains the unmodified P$^{tet}$tetA-T$^{hisL}$ construct. In strain MA14752 a 189 bp fragment encompassing phage Mu SGS is inserted mid-way between tetA and hilD. Strain MA14793 carries a mutant SGS (mut2, denoted by an asterisk) with sequence changes at the position of the gyrase cleavage site (see text and Supplementary Fig. 5 for details). **b** Percentage of cells expressing hilA-GFP$^{SF}$ in n = 4 independent repeats of the analysis in (**a**). **c** Quantification of hilD mRNA levels from the cultures used in (**b**). RNA from n = 4 or n = 5 independent experiments was quantified by RT-qPCR as described in the legend to Fig. 1. **d** Representative flow cytometry profiles from cultures of strains MA14696, MA14755 and MA14756 (left, middle and right panel, respectively) grown as in (**a**). Strain MA14696 carries a P$^{tet}$cat-T$^{hisL}$ cassette at the exact same position as the constructs in (**a**). Strains MA14755 and MA14756 are derivatives of MA14696 carrying wt SGS and a mutant SGS (mut1, denoted by an asterisk), respectively (see text and Supplementary Fig. 5 for details). **e** Percentage of cells expressing hilA-GFP$^{SF}$ in n = 4 independent repeats of the analysis in (**d**). **f** Quantification of hilD mRNA levels from the cultures used in (**e**). RNA from n = 3 or n = 4 independent experiments was quantified as described in (**c**). The error bars in (**b**, **c**, **e**, **f**) represent mean values ± SD. Statistical significance was determined by one-way ANOVA with either Šidák's (**b**, **e**) or Tukey's (**c**, **f**) multiple comparisons test. Adjusted P values were 0.029 and 0.052 in (**b**) (-SGS/+SGS and +SGS/+SGS* comparisons, respectively), 0.0005, 0,0931 and 0.0224 in (**c**) (-SGS/+SGS, -SGS/+SGS* and +SGS/+SGS* comparisons, respectively), 0.014 and 0.034 in (**e**) (-SGS/+SGS and +SGS/+SGS* comparisons, respectively) and <0.0001, 0.0016 and <0.0001 in (**f**) (-SGS/+SGS, -SGS/+SGS* and +SGS/+SGS* comparisons, respectively); (ns, P > 0.05; *P ≤ 0.05; **P ≤ 0.01; ***P ≤ 0.001; ****P ≤ 0.0001). Source data are provided as a Source Data file.

Concomitantly, the SGS mutations caused a substantial increase in hilD mRNA levels in both constructs (Fig. 4c, f). We interpret these findings to indicate that the SGS mutations impair the ability of gyrase to relax the transcription-induced positive supercoils. As a consequence, these supercoils migrate to the hilD promoter region, triggering the hilD self-activation loop.

## A DNA gyrase mutation abolishes bistability in hilA expression

We aimed to determine if mutations in DNA gyrase could function similarly to the SGS mutations in counteracting the effects of the SGS insertion. To explore this, we combined the tetA/SGS construct with two different gyrase alleles. The first allele, gyrA208, exhibits a DNA replication defect that may arise from the inefficient removal of positive supercoils accumulating in front of the replication fork[59,60]. Analyzing the strain carrying gyrA208 revealed a modest yet significant increase in the fraction of cells activating hilA-GFP in the SGS background during AHTc exposure (Fig. 5a, b). While the change is not as pronounced as that observed with the mutations in the SGS sequence, it does provide further support for the idea that inefficient removal of transcription-induced positive supercoils by gyrase enhances counter-silencing at the hilD promoter.

The second allele, gyrB1820, significantly impairs gyrase's ability to introduce negative supercoils into the DNA, as inferred from the extensive relaxation of reporter plasmids and reduced activity of supercoiling-sensitive promoters[59,61]. An intriguing picture emerged from the analysis of this allele. Cells containing gyrB1820 showed only minimal activation of hilA-GFP in response to AHTc treatment. However, the distribution of cell responses remained unimodal throughout the AHTc treatment, suggesting that the response is no longer bistable (Fig. 5c, d). This implies that negative DNA supercoiling is necessary for effective hilA (and presumably hilD) activation and that it might also play a role in the mechanism generating bistability.

## Discussion

Regulating the expression of virulence genes through H-NS silencing/counter-silencing mechanisms is recurrent among many species of free-living bacteria. H-NS-mediated regulation empowers pathogenic bacteria to swiftly adapt to the host environment, by fine-tuning the expression of functions that would otherwise prove detrimental or even toxic for growth outside the host[62]. It is increasingly evident that successful host adaptation hinges on an incomplete commitment to the virulence regulatory program. This incomplete commitment

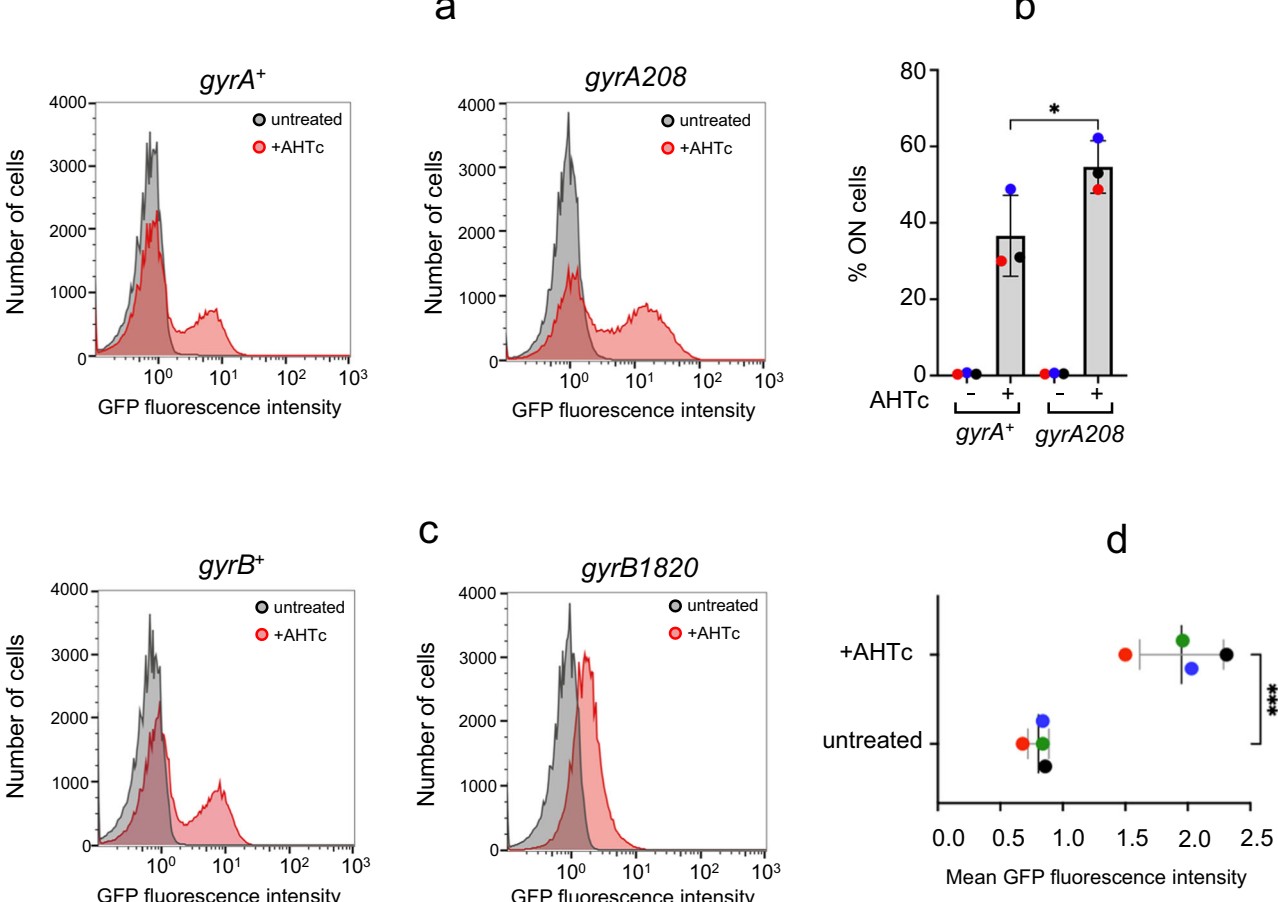

**Fig. 5 | Effects of gyrase mutations on *tetA*-mediated counter-silencing.**
**a** Representative flow cytometry profiles from cultures of strains MA14805 (P$^{tet}$*tetA*-T$^{hisL}$-SGS *gyrA$^+$*) and MA14804 (P$^{tet}$*tetA*-T$^{hisL}$-SGS *gyrA208*) grown to early stationary phase in the presence or absence of AHTc. The two strains are isogenic siblings from a transductional cross in which *gyrA208* was introduced in the P$^{tet}$*tetA*-T$^{hisL}$-SGS *hilA*-GFP$^{SF}$ background using a *gyrA*-linked *cat* insertion. **b** Percentage of cells expressing *hilA*-GFP$^{SF}$ in $n = 3$ independent repeats of the analysis in (**a**). **c** Representative flow cytometry profiles from cultures of strains MA14813 (P$^{tet}$*tetA*-T$^{hisL}$-SGS *gyrB$^+$*) and MA14812 (P$^{tet}$*tetA*-T$^{hisL}$-SGS *gyrB1820*) grown as in (**a**). The two strains are isogenic siblings from a transductional cross in which *gyrB1820* was introduced in the P$^{tet}$*tetA*-T$^{hisL}$-SGS *hilA*-GFP$^{SF}$ background using a *gyrB*-linked *cat* insertion. **d** Maximal GFP fluorescence values measured in *gyrB1820* mutant cells grown as in (**a**) in $n = 4$ independent experiments. Data in (**b**, **d**) are presented as mean values ± SD. Statistical significance was determined by RM one-way ANOVA with Šidák's multiple comparisons test in (**b**) (adjusted $P$ value for the *gyrA$^+$*/*gyrA208* comparison = 0.0188) and the unpaired two-tailed Student's $t$ test in (**d**) ($P = 0.0006$); (*$P \le 0.05$; ***$P \le 0.001$). Source data are provided as a Source Data file.

results in the coexistence, within the bacterial population, of cells that either express or do not express this program[34,63]. Defining the exact role of H-NS in the mechanisms that generate bistability in expression patterns should shed light on the evolutionary dynamics of pathogenicity[64,65].

Here we presented strong evidence that transcription, when coupled with translation, effectively alleviates H-NS-mediated gene silencing through a long-range mechanism. This transcription-induced counter-silencing can be partially or completely suppressed by recruiting DNA gyrase to the region that separates the transcription unit from the silenced gene. This leads us to conclude that positive DNA supercoils generated during transcription elongation, especially when the elongation complex is associated with the leading ribosome, play a crucial role in counter-silencing. If not resolved by gyrase, these positive supercoils migrate towards the silenced gene, ultimately activating it, likely by disrupting the H-NS:DNA complex. A direct implication of this scenario is that positive DNA supercoiling destabilizes the H-NS bindings to its DNA targets.

In retrospect, H-NS's aversion to positively supercoiled DNA may not be surprising. H-NS has long been known to constrain negative DNA supercoiling both in vivo and in vitro[66,67]. Resolution of the crystal structure of an 83-unit H-NS oligomer revealed that the molecule

adopts a right-handed spiraling conformation suitable for tracking the contours of a negatively supercoiled DNA plectoneme[68]. In that study, authors proposed that H-NS filaments might serve as a scaffold for negatively supercoiled DNA. Collectively, these considerations suggest a model for how positive supercoiling could promote the disassembly of H-NS:DNA complexes. Our model (Fig. 6) posits that H-NS-bound DNA forms a nucleoprotein complex where H-NS bridges the opposite arms of a negatively supercoiled plectoneme. Positive supercoils generated during the transcription of an upstream gene would diffuse toward the complex through axial rotation and merge with the negatively supercoiled H-NS-bound DNA[69,70]. We propose that the rotation of the DNA axis will effectively unroll the negatively supercoiled H-NS-bound plectoneme, causing H-NS protein units to dissociate.

The model above can explain not only the long-range effects of transcription on neighboring H-NS:DNA complexes, as described in this paper, but it is also relevant to the mechanism by which transcription elongates through a H-NS-bound region. In the latter scenario, an intriguing possibility is that H-NS:DNA complexes disrupted ahead of the RNA polymerase rapidly reassemble after the passage of the elongation complex by binding to the negatively supercoiled domain that forms behind the polymerase. This process can be expected to depend on the level of transcription, since close spacing

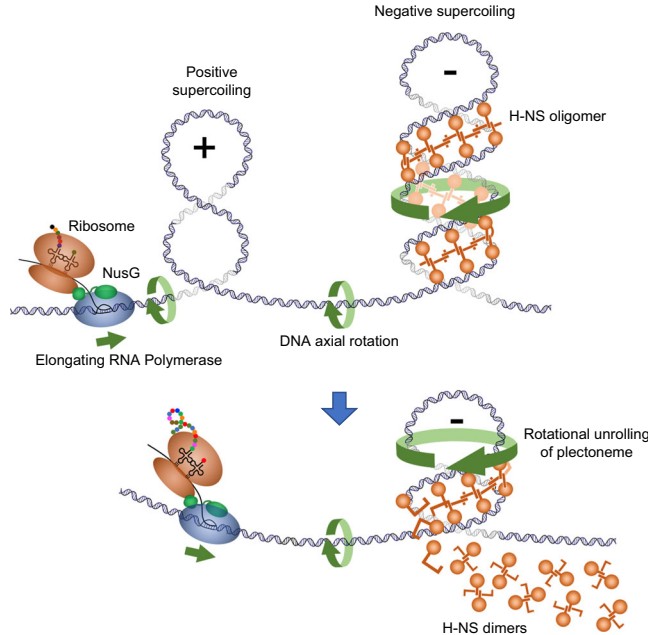

**Fig. 6 | Model for how transcription-driven positive DNA supercoiling may cause the disassembly of H-NS:DNA complexes.** The model posits that H-NS:DNA complexes exist in vivo in the form of H-NS-bound negatively supercoiled DNA. The H-NS filament follows the path of the righthanded plectoneme bridging sequences from its two arms. During transcription of a neighboring gene, frictional resistance to the rotation of the transcription complex around the DNA axis forces the DNA axis to rotate on itself, thereby generating positive supercoils. This effect is amplified by the association of the leading ribosome with the elongation complex and even more so by the co-transcriptional anchoring of a nascent polypeptide to the cell membrane (see text). If not relaxed by DNA gyrase, positive supercoils diffuse along the DNA and merge with the H-NS-bound negatively supercoiled plectoneme. Rotation of the DNA axis causes the plectoneme to unroll. Unrolling breaks the H-NS bridges resulting in the release of H-NS dimers.

between polymerases in highly transcribed genes may cause positive and negative supercoils between polymerases to mutually cancel out[71]. This overall scenario aligns with observations from ChIP experiments where substantial amounts of H-NS remain bound to target genes even when these genes are actively transcribed[22,31,72] (see also Fig. 2a). Transcriptional supercoiling may also support RNA polymerase progression through regions bound by other nucleoid-associated proteins, for example DPS, which compacts the DNA through oligomerisation of DPS dodecamers. Despite forming phase-separated condensates, DPS:DNA complexes remain permeable to RNA polymerase and maintain the dynamic condensed structure during RNA chain elongation[73].

The idea that chromatin components could dynamically translocate from the front to the rear of RNA polymerase during transcription was originally proposed for histone octamers in eukaryotes[74]. Note that the left-handedness of the toroidal wrapping of DNA around the histone core predicts that positive DNA supercoiling should destabilize nucleosomes. Consistent with this prediction, various lines of evidence, both in vitro and in vivo, indicate that transcription-driven positive supercoiling causes nucleosomes to disassemble, and that this is potentially important for transcription to elongate through eukaryotic chromatin[75–80]. Extending the analogy with our data one step further, one might speculate that long-range effects of the type described here could be used to destabilize nucleosomes in promoter regions so as to regulate transcription initiation by eukaryotic RNA polymerases.

The findings described in this paper raise a further question on the possible general role of positive DNA supercoiling in H-NS counter-

silencing mechanisms. A recent study showed that the VirB protein of *Shigella* relieves the H-NS silencing of virulence plasmid genes through the generation of positive supercoils in the plasmid DNA[81]. In this system, positive supercoiling is generated locally as a result of VirB protein binding to plasmid DNA. VirB binds outside the H-NS-bound region, which is itself separated from the silenced promoter. VirB forms filaments along the DNA, similarly to what was initially described for proteins of the ParB family of which VirB could be a member[82]. Importantly, ParB family proteins also induce positive supercoils in DNA, implying that oligomerisation is central to their ability to affect DNA topology[83,84]. However, outside the VirB-bound region, the DNA becomes more negatively supercoiled (not positively) as evidenced by the recruitment of topoisomerase I[81]. One would have to conclude that unconstrained positive supercoils are generated transiently by some unknown activity of the VirB:DNA complex.

In *Salmonella*, HilD is thought to function as both an H-NS counter-silencer and a classical transcriptional activator, blurring the boundary between the two functions[36,85,86]. Comparing HilD and H-NS genomic DNA binding profiles in ChIPSeq experiments[31,87] reveals that the HilD binding sites fall inside regions bound by H-NS and in close proximity to HilD/H-NS regulated promoters. Combining the lack of evidence for HilD filament formation on DNA and the present observations, it seems likely that HilD and similar dual-function activators work differently from VirB. Thus, at first glance, it is difficult to envisage a role for DNA supercoiling in the functioning of HilD, aside from the response to transcription-driven supercoiling[31]. Nonetheless, it is intriguing that under conditions where overall negative superhelicity of chromosomal DNA is greatly reduced due to the *gyrB1820* mutation, HilD appears to lose its ability to fully activate *hilA*-GFP[SF] expression (Fig. 5c). Even more intriguingly, the low-level activation now occurs homogeneously in the whole cell population; that is, *gyrB1820* abrogates bistability. This might imply that normal negative supercoiling levels are required not only for the proper functioning of the transcriptional activator but also for the proper functioning of the mechanism that keeps a fraction of the population in the OFF state under activating conditions. These findings open a new perspective to the study of bistability and provide a new avenue for investigating the functional dynamics of bacterial chromatin.

## Methods
### Strains and culture conditions
All strains used in this work are derived from *Salmonella enterica* serovar Typhimurium strain LT2[88]. Strains and their genotypes are listed in Supplementary Table 1. Bacteria were routinely cultured in Lysogeny Broth (LB) (Lennox formulation: Tryptone 10 g/l, Yeast extract 5 g/l, NaCl 5 g/l) at 37 °C or, occasionally, at 30 °C when carrying temperature-sensitive plasmid replicons. Typically, bacteria were grown overnight in static cultures, subcultured by a 1:200 dilution the next day and grown with 170 rpm shaking. For growth on plates, LB was solidified by the addition of 1.5% Difco agar. When needed, antibiotics (Sigma-Aldrich) were included in growth media at the following final concentrations: chloramphenicol, 10 μg/ml; kanamycin monosulphate, 50 μg/ml; sodium ampicillin 100 μg/ml; spectinomycin dihydrochloride, 80 μg/ml; tetracycline hydrochloride, 25 μg/ml. Strains were constructed by generalized transduction using the high-frequency transducing mutant of phage P22, HT 105/1 *int-201*[89] or by λ-*red* recombineering, following a standard protocol[90] or a protocol producing seamless modifications[91]. The construction of relevant strains is described in Supplementary Methods. Oligonucleotides used as primers for the amplification of the recombineering fragments are listed in Supplementary Table 2. PCR amplification was with high-fidelity Phusion polymerase (New England Biolabs). Constructs were verified by colony-PCR using Taq polymerase followed by DNA sequencing (performed by Eurofins-GATC Biotech).

## Fluorescence microscopy

Bacterial cultures grown overnight in LB at 37 °C were diluted 1:200 in the same medium with or without 0.1% arabinose (ARA) and/or 0.4 μg/ml anhydrotetracycline (AHTc) and grown for 4 h at 37 °C with shaking (170 rpm). Cells were then harvested by centrifugation (2 min at 12,000 × $g$), washed once in PBS and used immediately for microscopic examination. Images were captured with a Leica DM 6000 B microscope (CTR 6500 drive control unit) equipped with a EBQ 100 lamp power unit and filters for phase contrast, GFP and mCherry detection (100 x oil immersion objective). Pictures were taken with a Hamamatsu C11440 digital camera and processed with Metamorph software.

## Flow cytometry

Flow cytometry was used to monitor expression of the *hilA*-GFP$^{SF}$ fusions. Data acquisition was performed using a Cytomics FC500-MPL cytometer (Beckman Coulter, Brea, CA) and data were analyzed with FlowJo X version 10.0.7r software (Tree Star, Inc., Ashland, OR). *S. enterica* cultures were washed and re-suspended in phosphate-buffered saline (PBS) for fluorescence measurement. Fluorescence values for 100,000 events were compared with the data from the reporterless control strain, thus yielding the fraction of ON and OFF cells. The gating strategy is exemplified in Supplementary Fig. 6.

## RNA extraction and quantification by RT-qPCR

Overnight bacterial cultures in LB were diluted 1:200 in the same medium - or in LB supplemented with 0.4 μg/ml AHTc or 0.1% ARA or both - and grown with shaking at 37 °C to an $OD_{600}$ = 0.7 to 0.8. Cultures (4 ml) were rapidly spun down and resuspended in 0.6 ml ice-cold REB buffer (20 mM Sodium Acetate pH 5.0, 10% sucrose). RNA was purified by sequential extraction with hot acid phenol, phenol-chloroform 1:1 mixture and chloroform. Following overnight ethanol precipitation at −20 °C and centrifugation, the RNA pellet was resuspended in 20 μl of $H_2O$. Samples were prepared from three to five independent biological replicates for each strain or condition. RNA yields, measured by Nanodrop or Qubit reading, typically ranged between 2 and 2.5 μg/μl. The RNA preparations were used for first-strand DNA synthesis with the New England Biolabs (NEB) ProtoScript II First Strand DNA synthesis kit, following the manufacturer's specifications. Briefly, RNA (1 μg) was combined with 2 μl of a mixture of two primers (5 μM each), one annealing in the promoter-proximal portion of the RNA to be quantified, the other annealing to a similar position in the RNA used for normalization (typically *ompA* mRNA) in an 8-μl final volume. After 5 min at 65 °C and a quick cooling step on ice, volumes were brought to 20 μl by the addition of 10 μl of ProtoScript II Reaction Mix (2x) and 2 μl of ProtoScript II Enzyme Mix (10x). Mixes were incubated for one hour at 42 °C followed by a 5 min enzyme inactivation step at 80 °C. Samples were then used for real time quantitative PCR. PCR reactions were set up in 384-well plates by mixing serial dilutions of each reverse-transcribed sample with the appropriate primer pairs (each primer used at a 0.25 μM final concentration) and the LightCycler 480 SYBR Green I Master Mix (Roche Applied Science). Real-time qPCR was carried out in a LightCycler 480 Instrument (Roche) with the following program: activation: 95 °C for 5 min; amplification (40 cycles): 95 °C for 10 s; 55 °C for 20 s; 72 °C for 20 s; melting curve: 95 °C for 30 s; 65 °C for 30 s (ramp 0.06 °C/s, 10 acquisitions/°C). Target-to-reference transcript ratios and relative transcript levels were calculated with the Pflaffl method[92]. The oligonucleotides used as primers in the RT and qPCR steps are listed in Supplementary Table 3.

## 5′ RACE-Seq analysis

RNA 5′-end analysis was performed by template switching reverse transcription as recently described[31]. Briefly, total RNA (1 μg) was combined with 1 μl of a mixture of up to three gene-specific primers, 5 μM each (including AI69 (*hilD*), AI48 (*prgH*) and AJ33 (*ompA*)), and 1 μl

of 10 mM dNTPs in a 6 μl final volume. After a 5 min treatment at 70 °C (in a Thermocycler), samples were quickly cooled on ice. Each sample was then mixed with 2.5 μl of Template Switching Buffer (4x), 0.5 μl of 75 μl of Template Switching Oligonucleotide (TSO) and 1 μl of Template Switching RT Enzyme Mix in a final volume of 10 μl. Reverse transcription was carried out for 90 min at 42 °C, followed by a 5 min incubation at 85 °C. The synthesized cDNA was amplified by PCR with primers carrying Illumina adapters at their 5′ ends. Several PCRs were carried out in parallel with a common forward primer (AJ38, annealing to the TSO) and a reverse primer specific for the region being analysed and carrying a treatment-specific index sequence. Reactions were set up according to New England Biolabs PCR protocol for Q5 Hot Start High-Fidelity DNA polymerase in a final volume of 50 μl (using 1 μl of the above cDNA preparation per reaction). The number of amplification cycles needed for reproducible semiquantitative measurements, determined in trial experiments, was chosen to be 25 for the *ompA* promoter, 30 for the primary *hilD* and *prgH* promoters and 35 for the *leuO* promoter. The PCR program was as follows: activation: 98 °C for 30 s; amplification (25 or 30 or 35 cycles): 98 °C for 10 s; 65 °C for 15 s; 72 °C for 30 s; final stage: 72 °C for 5 min. The PCR products were mixed in equal volumes; mixes originating from the amplification of separate regions were pooled and the pools subjected to high throughput sequencing. The procedure was implemented at least once, occasionally twice, with each of the independent RNA preparations. The counts of reads containing the TSO sequence fused to the TSS of interest, each normalized to the counts of reads containing the TSO fused to the reference TSS, were used to calculate the ratios between the activity of a promoter following AHTc exposure relative and its activity in untreated cells (for more information on the design of the experiments and the primers used, see Supplementary Methods and Supplementary Table 4).

## ChIP-Seq analysis

ChIP-Seq analysis was conducted as previously described[31]. Raw and processed data were deposited into ArrayExpress under the accession number E-MTAB-13436.

## Statistics and reproducibility

All data described in this paper originate from three or more independent experiments, with one or more measurements performed on each replicate of the experiment. Statistical significance was calculated as specified in the legends to the figures. All statistical analyses were done GraphPad Prism 9 software. $P$ values were included in the figures (using the asterisk symbol) or specified in the figure legends.

## Bioinformatic analyses

Demultiplexing of raw data from the Illumina sequencer was performed with the bcl2fastq2 V2.2.18.12 program and adapters were trimmed with Cutadapt1.15. The reads from the ChIP-Seq experiments were mapped on the genome of *Salmonella enterica* serovar Typhimurium strain MA14443 (SPI-1-modified derivative of strain LT2) with BWA 0.6.2-r126. Bedgraph files were generated from aligned Bam files using bedtools genomecov. Coverage track (number of reads per base) was converted to the BigWig format using the bedGraphToBigWig command line utility from UCSC. Read counts in selected regions were calculated using the Samtools view tool of the Samtools suite. Bam and Bigwig files were visualized with Integrative Genome Viewer (IGV)[93]. In the processing of the RACE-Seq data, the reads containing the TSO were filtered and kept thanks to grep command in the SeqKit package. These reads were then trimmed with the PRINSEQ tool to remove 33 bp from the 5′ end (the amplified portion of the TSO except the terminal 3Gs).

## Reporting summary

Further information on research design is available in the Nature Portfolio Reporting Summary linked to this article.

## Data availability

The ChiP-Seq data generated in this study have been deposited in the ArrayExpress database under the accession code E-MTAB-13436. The FASTA and gff3 files of *Salmonella enterica* serovar Typhimurium strain MA14443 used in the ChiP-Seq analysis are provided in the Source Data file. The 5′RACE-Seq data in Fig. 3c and Supplementary Fig. 4a have been deposited in the ArrayExpress database under the accession code E-MTAB-13482. The 5′RACE-Seq data in Supplementary Fig. 4b are provided in the Source Data file. The density plots from the flow cytometry analysis in Fig. 1a–d; Fig. 4a, d and Fig. 5a, c are provided in the Source Data file. The raw data from the experiments in Fig. 1e, f, g; Fig. 2b; Fig. 3b, d; Fig. 4b, c, e, f; Fig. 5b, d; Supplementary Fig. 3b, and Supplementary Fig. 4c, d are provided in the Source Data file. Source data are provided with this paper.

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

## Acknowledgements

We thank Alexandra Gruss for critical reading of the manuscript. We also thank the High-throughput Sequencing Core Facility of the I2BC (Gif-sur-Yvette, France) for library preparation and sequencing services. This work was funded by the Centre National de la Recherche Scientifique (CNRS) France, by the Agence Nationale de la Recherche, France (ANR-15-CE11-0024-03), and by the grant PID2020-116995RB-I00 funded by MCIN/AEI/10.13039/5011100011033 (Ministerio de Ciencia e Innovación, Spain), and, as appropriate, by "ERDF A way of making Europe", or by the "European Union" or by the "European Union Next Generation EU/PRTR" and the VI Plan Propio de Investigación y Transferencia from the University of Seville.

## Author contributions

N.F.B., M.A.S.R., J.C. and L.B. conceived and initiated this study. P.K. provided technical support, performed strain construction and verifications, analysed data. M.A.S.R and R.F.F. designed, obtained data and analysed single cell experiments, prepared figures and tables. P.B. provided valuable intellectual input throughout the study and critically reviewed the manuscript. N.F.B. and L.B. coordinated the study, designed, performed experiments and analysed data, prepared and edited figures and tables L.B. wrote the article with the help of N.F.B.

## Competing interests

The authors declare no competing interests.
