## [Peer Review File · Nature Communications]

Transcription-driven DNA supercoiling counteracts H-NS-mediated gene silencing in bacterial chromatinReviewer #1 (Remarks to the Author):

The manuscript titled "Transcription-driven DNA supercoiling activates bacterial chromatin at a distance" offers a new perspective on gene regulation in bacteria. While gene activation by topological stress is well-documented and widely studied in eukaryotic cells, little is known about a similar process in prokaryotic organisms, and this work provides a fresh insight into this level of gene expression. The experimental workflow is logical, concise, and well-written. However, there are some issues that should be addressed to further improve the quality of the publication.

1. To demonstrate that *hldD* is activated due to torsional stress and not via direct transcription from the previous gene's promoter, the authors have compromised translation from the upstream *tetA* RNA and observed that the activation is lost. While the authors briefly mention that the disappearance of activation is due to premature termination by the Rho factor, this should be proven separately. This is because mRNA bound to ribosomes may provide a greater hindrance for RNA polymerase axial rotation than naked RNA. This can be achieved easily by using either bicyclomycin or a temperature-sensitive Rho mutant to turn off Rho activity. If transcription alone is sufficient to provide torsional stress and translation only protects from premature termination, then turning off Rho activity should activate downstream *hldD*, regardless of mutations in the initiator codon.

2. When the authors study if activation via torsion stress is a locus-specific or more general phenomenon, they insert the *tetA-ThisL* construct upstream of the *leuO* gene and observe an effect similar to *SPL-1*. To ensure that this effect is specific, it is necessary to show that the upstream and divergent *LeuL* or *LeuA* genes are not activated upon firing the *tetA* promoter. This can be easily measured via RT-qPCR.

Minor:

1. It would be helpful to indicate the precise location of SGS between *tetA* and *hldD*, particularly with regard to the position of the *hisL* terminator.

2. The authors discuss transcription regulation via torsional stress among different kingdoms of life but fail to mention the potential effects of such stress on other DNA-packaging proteins in bacteria besides H-NS, such as DPS.

3. In lines 79-81, please provide a citation: PMID: 18487194.

4. In lines 86-87, please include a citation: PMID: 33296676.

Reviewer #2 (Remarks to the Author):

In this paper Figueroa-Bossi and co-workers build on their recent discovery that spurious transcription initiation can counter H-NS mediated silencing of pathogenicity islands. In this work, the authors build on their prior observations to show that, when coupled to translation, transcription at a distance is able to derepress H-NS mediated silencing. It is proposed that this effect is mediated by diffusion of positive supercoils generated by the elongating RNA polymerase. To test this model, the authors introduce a DNA gyrase binding site between the site of transcription initiation and H-NS silenced region. The expectation is that the actions of DNA gyrase will prevent supercoil diffusion and thus the observed counter silencing. Overall, I thought the work was beautifully explained and presented.

A difficulty the authors face is that direct detection of supercoiling changes *in vivo*, at sufficient resolution, is not possible with current technology. As such, they are reliant on the indirect assumption that supercoil diffusion is blocked by the SGS sequence. I like the experiment but also wonder about alternative explanations. Hence, it would be useful if the authors could comment on the likelihood of inadvertent effects due to the SGS sequence. In particular, is this sequence expected to be transcribed and translated? If so, are there in frame stop codons that could have an

impact? Also, can the authors comment on the possibility that transcription, perhaps in the direction of tetA, could initiate within the SGS sequence? This might interfere with supercoiling driven by tetA expression.

A potential criticism is that the authors assume, but do not show, that positive supercoiling disrupts H-NS binding. Overall, I do buy into the explanation the authors provide in the discussion, referring to work from John Ladbury's group. That said, it would perhaps not be very difficult to show effects of positive supercoiling on DNA binding by H-NS in vitro (e.g. gel shift assays before and after treatment of plasmids with reverse gyrase).

In summary, I thought this was a nice story and I'm supportive of the work.

Reviewer #3 (Remarks to the Author):

The SPI1 regulatory gene, *hilD*, is silenced by H-NS and autoactivated by HilD. This contributes to the bistable expression of SPI1 and provides a sensitive system to study H-NS silencing and the factors that affect this silencing. These authors have previously shown that spurious transcription at a distance from the *hilD* promoter can lead to activation. Here, they provide a more mechanistic understanding of this effect consistent with changes in supercoiling caused by transcription and translation leading to disruption of H-NS binding at a distance. This paper is an important contribution to our understanding of H-NS DNA interaction and the factors that affect this interaction. This work is relevant to many Gram-negative bacteria and likely to other nucleoid proteins.

1. Title: It is not clear to me that "...activates bacterial chromatin..." has any agreed upon meaning. I would consider a new title.

2. The paper begins by covering the data in two supplementary figures. Adding the corresponding 1.2 kb tetR ptet but tetA minus strain to Fig 1 would provide a more complete set and relieve the casual reader from having to go through supplementary figure. This is just a suggestion.

3. Supplementary Fig 4. This fig could be labeled *hilD* and *hilA* rather than just a and b, which forces the reader to study the legend. Additional figures could also be enhanced with more labels.

4. Supplementary Fig 4. More importantly, published data suggest that activation of the *prgH* promoter is strictly dependent on HilA, even when H-NS binding is disrupted. Therefore, it is surprising that transcription/translation of tetA seems to induce *prgH* transcription in a strain (MA14692) that is *hilA*'-'GFP (*hilA* null). The authors should note that SPI1 is readily duplicated by an unknown mechanism. These duplications are likely induced or created during transduction. This has previously led to misinterpretation of data (eg Murray and Lee. 2000. Infect Immun). The authors should ensure that this strain does indeed lack a wildtype copy of *hilA*.

5. Line 221. "Introduction of the SGS led to a significant..."

6. Line 224. The authors switch Tet with Cat to test the hypothesis that translation of the TetA membrane protein adds to the supercoiling effect. A caveat is that the GC content of the cat gene is much lower than tetA. The authors should think about how this could affect the interpretation of these results and perhaps point it out to the reader.

7. Figures. The color choices for some of the figures make it difficult to distinguish the differences. The authors should maximize contrast and use different colors only when it is necessary to understand the experiment.

Reply to Reviewers

Reviewer comments in italics; our reply in plain text.

REVIEWER COMMENTS

Reviewer #1 (Remarks to the Author):

The manuscript titled "Transcription-driven DNA supercoiling activates bacterial chromatin at a distance" offers a new perspective on gene regulation in bacteria. While gene activation by topological stress is well-documented and widely studied in eukaryotic cells, little is known about a similar process in prokaryotic organisms, and this work provides a fresh insight into this level of gene expression. The experimental workflow is logical, concise, and well-written. However, there are some issues that should be addressed to further improve the quality of the publication.

*1. To demonstrate that *hilD* is activated due to torsional stress and not via direct transcription from the previous gene's promoter, the authors have compromised translation from the upstream *tetA* RNA and observed that the activation is lost. While the authors briefly mention that the disappearance of activation is due to premature termination by the Rho factor, this should be proven separately. This is because mRNA bound to ribosomes may provide a greater hindrance for RNA polymerase axial rotation than naked RNA. This can be achieved easily by using either bicyclomycin or a temperature-sensitive Rho mutant to turn off Rho activity. If transcription alone is sufficient to provide torsional stress and translation only protects from premature termination, then turning off Rho activity should activate downstream *hilD*, regardless of mutations in the initiator codon.*

To turn off Rho activity, we use a strain in which the *nusG* gene is under the control of an arabinose-inducible repressor. Exposure to arabinose (ARA) causes cells to become depleted for NusG, which in turn causes Rho inhibition throughout the pathogenicity island (PMID: 31589608). Note that Rho inhibition *per se* already activates *hilD* expression due to runaway transcription originating from spurious promoters scattered in the regions neighbouring the *hilD* gene (PMID: 35858437). To determine if *tetA* transcription adds to these effects in the absence of translation, and does so at a distance (thus presumably via supercoiling), we first had to combine the *tetA* initiator codon mutation (AUG > AAA) with the Rho-independent *hisL* terminator at the end of *tetA* (in the construct with the *tetA*-T^{hisL} cassette positioned 1.2 Kb from *hilD*). We then moved the construct into the NusG-depletable background and measured *hilD* mRNA levels in cells treated with either AHTc, ARA or these two products combined. Results confirmed AHTc alone to have only a modest two-fold effect, whereas ARA led to an approximately 13-fold increase in *hilD* mRNA levels. However, in cells treated with ARA and AHTc combined, the increase was more than 60-fold. The 4.5-fold difference between the ARA-treated samples with or without AHTc is thus entirely attributable to untranslated transcripts that initiate at P^{tet} and elongate up to the *hisL* terminator. It is a relatively small effect when compared to the 100-fold increase of *hilD* mRNA when *tetA* is translated (Fig; 1b, c), but

nonetheless significant as it **implies that translation, while amplifying supercoil partitioning, is not absolutely required for transcription to generate supercoils**. We have included these results in the revised version of the manuscript (**panel g in Fig. 1**; described in lines **158-177** in the text and in lines **808-813** in Fig. 1 legend). We thank the reviewer for suggesting to perform this analysis.

2. When the authors study if activation via torsion stress is a locus-specific or more general phenomenon, they insert the tetA-ThisL construct upstream of the leuO gene and observe an effect similar to SPL-1. To ensure that this effect is specific, it is necessary to show that the upstream and divergent LeuL or LeuA genes are not activated upon firing the tetA promoter. This can be easily measured via RT-qPCR.

Unfortunately, we could not perform this test. To avoid possible interference from the *leuLABCD* promoter, the *tetA-T^{hisL}* construct was made by concomitantly deleting a 112 bp fragment encompassing this promoter (note the little gap between the left and right junctions of the insert upstream of the *leuL* orf in Fig. 3A). We have now specified that the *leuLABCD* promoter is deleted in the text (lines **220-222**).

Minor:

1. It would be helpful to indicate the precise location of SGS between tetA and hild, particularly with regard to the position of the hisL terminator.

A diagram indicating the position of the SGS insert in relative to the *hisL* terminator and the *hild* promoter has been added to **Supplementary Fig. 5 (panel a)**. The information is also provided in writing in the text (lines **250-252**). Note also the labels added to Fig. 4 in response to point 3 of Reviewer n. 3.

2. The authors discuss transcription regulation via torsional stress among different kingdoms of life but fail to mention the potential effects of such stress on other DNA-packaging proteins in bacteria besides H-NS, such as DPS.

We now mention the possibility that transcriptional supercoiling helps displace DPS during transcription of DPS:DNA condensates (lines **351-355**).

3. In lines 79-81, please provide a citation: PMID: 18487194.

We have added the citation (line **83**). However, please note that our text here is specifically focused on the interplay between Rho and H-NS in the silencing of foreign DNA. While the seminal paper by Cardinale and coworkers provided the first demonstration for the role of Rho/NusG in silencing foreign DNA, it failed to incorporate the Rho:H-NS interplay into the picture. H-NS is briefly mentioned only once at the very end of the paper and described as a separate “immunity” system.

4. In lines 86-87, please include a citation: PMID: 33296676.

We have now added the citation together with a new sentence that clarifies its pertinence in this context (lines **89-91**).

Reviewer #2 (Remarks to the Author):

In this paper Figueroa-Bossi and co-workers build on their recent discovery that spurious transcription initiation can counter H-NS mediated silencing of pathogenicity islands. In this work, the authors build on their prior observations to show that, when coupled to translation, transcription at a distance is able to derepress H-NS mediated silencing. It is proposed that this effect is mediated by diffusion of positive supercoils generated by the elongating RNA polymerase. To test this model, the authors introduce a DNA gyrase binding site between the site of transcription initiation and H-NS silenced region. The expectation is that the actions of DNA gyrase will prevent supercoil diffusion and thus the observed counter silencing. Overall, I thought the work was beautifully explained and presented.

A difficulty the authors face is that direct detection of supercoiling changes in vivo, at sufficient resolution, is not possible with current technology. As such, they are reliant on the indirect assumption that supercoil diffusion is blocked by the SGS sequence. I like the experiment but also wonder about alternative explanations. Hence, it would be useful if the authors could comment on the likelihood of inadvertent effects due to the SGS sequence. In particular, is this sequence expected to be transcribed and translated? If so, are there in frame stop codons that could have an impact? Also, can the authors comment on the possibility that transcription, perhaps in the direction of tetA, could initiate within the SGS sequence? This might interfere with supercoiling driven by tetA expression.

The SGS sequence contains multiple stop codons in all six reading frames. None of the SGS mutants that restore supercoil diffusion eliminates the translational block in any one frame, making it unlikely that supercoil diffusion is in any way related to the lack of translatability of the sequence. Regarding the possible presence of a promoter pointing toward *tetA* within the SGS sequence, we have now tested this by measuring *prgH* RNA levels (by RT-qPCR) in the segment between the *tetA* gene and the SGS position in the strains with and without the SGS insert. This analysis, performed in cells growing in the presence or absence of AHTc, revealed that the RNA levels in the strain carrying the SGS **are actually lower** (~10-fold in minus AHTc; ~50-fold in plus AHTc) than in the parental strain lacking the SGS under the same conditions. This finding **rules out the presence of a promoter in the SGS sequence** and suggests that the RNA measured in this experiment originates entirely from the *prgH* promoter. The lower level of this RNA in the SGS strain compared to the parental strain is likely attributable to polarity. In the parental strain, the *prgH* mRNA is normally translated up to the *tetA* insert; in the SGS strain, translation is interrupted by the stop codons in the SGS insert. This interruption may cause the RNA to be prematurely terminated and/or degraded.

A potential criticism is that the authors assume, but do not show, that positive supercoiling disrupts H-NS binding. Overall, I do buy into the explanation the authors provide in the discussion, referring to work from John Ladbury's group. That said, it would perhaps not be very difficult to show effects of positive supercoiling on DNA binding by H-NS in vitro (e.g. gel shift assays before and after treatment of plasmids with reverse gyrase).

We thank the reviewer for this suggestion. Unfortunately, this analysis would have required materials (purified H-NS and reverse gyrase) and expertise that are impossible for us to obtain within the time-frame of the revision process and would very much delay the publication of the current data.

In summary, I thought this was a nice story and I'm supportive of the work.

Reviewer #3 (Remarks to the Author):

The SPI1 regulatory gene, hilD, is silenced by H-NS and autoactivated by HilD. This contributes to the bistable expression of SPI1 and provides a sensitive system to study H-NS silencing and the factors that affect this silencing. These authors have previously shown that spurious transcription at a distance from the hilD promoter can lead to activation. Here, they provide a more mechanistic understanding of this effect consistent with changes in supercoiling caused by transcription and translation leading to disruption of H-NS binding at a distance. This paper is an important contribution to our understanding of H-NS DNA interaction and the factors that affect this interaction. This work is relevant to many Gram-negative bacteria and likely to other nucleoid proteins.

1. Title: It is not clear to me that "...activates bacterial chromatin..." has any agreed upon meaning. I would consider a new title.

The title was changed to "Transcription-driven DNA supercoiling counteracts H-NS-mediated gene silencing in bacterial chromatin". We believe that maintaining the term "chromatin" might attract a wider readership.

2. The paper begins by covering the data in two supplementary figures. Adding the corresponding 1.2 kb tetR ptet but tetA minus strain to Fig 1 would provide a more complete set and relieve the casual reader from having to go through supplementary figure. This is just a suggestion.

We thank the reviewer for this suggestion. The data from P^{tet}-tetA minus construct are now included in **Figure 1 (panel a)**. Panel **e** and the figure legend were modified accordingly. Note also the nomenclature change: "Construct 1, 2," etc were replaced by "Strain A, B" etc.

3. Supplementary Fig 4. This fig could be labeled hilD and hilA rather than just a and b, which forces the reader to study the legend. Additional figures could also be enhanced with more labels.

We have added the *hilD* and *hilA* labels (but left **a** and **b** for consistency with the other figures). Further labels, in the form of schematic diagrams and new lettering, were added to **Fig. 4** to facilitate the reading of this figure. The legend to Fig. 4 was modified accordingly.

*4. Supplementary Fig 4. More importantly, published data suggest that activation of the *prgH* promoter is strictly dependent on HilA, even when H-NS binding is disrupted. Therefore, it is surprising that transcription/translation of *tetA* seems to induce *prgH* transcription in a strain (MA14692) that is *hilA*'-GFP (*hilA* null). The authors should note that SPI1 is readily duplicated by an unknown mechanism. These duplications are likely induced or created during transduction. This has previously led to misinterpretation of data (eg Murray and Lee. 2000. Infect Immun). The authors should ensure that this strain does indeed lack a wildtype copy of *hilA*.*

The presence of a duplication containing wild-type *hilA* in one copy of the duplicated material was tested by PCR using a three-primer mixture, a forward primer annealing in the promoter-proximal region of *hilA* (present in the *hilA*'-GFP fusion) and two reverse primers one annealing to the GFP gene, the other to the distal portion of *hilA* (absent in the *hilA*'-GFP fusion). Presence of a duplication should result in the concomitant amplification of two fragments of distinct lengths. This was not the case with any of the representative group of strains used in this study. All these strains showed the expected single band; either the *hilA*⁺-specific fragment or the *hilA*'-GFP fusion-specific fragment. **We conclude that there is no duplication in our strains.**

Note that the *hilA*'-GFP fusion may not be an *hilA* null. This translational fusion retains the N-terminal 112 amino acids implicated in *prgH* promoter recognition, raising the possibility that the chimera is proficient in regulation (PMID: 35858437). To clarify this point, we have now analysed the effects of P^{tet} activation in a strain where *hilA* is completely deleted. Results clearly showed that *prgH* is strongly induced even in this background (to a level comparable to that observed with the *hilA*-GFP fusion). Thus, it appears that when triggered by upstream transcription, *prgH* activation, while completely dependent on HilD (PMID: 35858437), occurs independently of HilA. This information is now mentioned in the revised version (lines **188-195**).

5. Line 221. "Introduction of the SGS led to a significant..."

Corrected.

*6. Line 224. The authors switch Tet with Cat to test the hypothesis that translation of the TetA membrane protein adds to the supercoiling effect. A caveat is that the GC content of the *cat* gene is much lower than *tetA*. The authors should think about how this could affect the interpretation of these results and perhaps point it out to the reader.*

The GC content of the *cat* gene (44.85 %) is actually **slightly higher** than that of *tetA* (43.28%). We don't see how this difference could account for the difference in the supercoiling effect. On the other hand, an alternative explanation for the differential effect may lie in the size

difference between the two genes: *cat* being near half the size of *tetA* (657 vs 1203 bp), it could constitute a lower drag for RNA rotation around the DNA during transcription elongation and thus generate less supercoiling. A sentence mentioning this possibility has been included in the revised version (lines **259-261**).

7. Figures. The color choices for some of the figures make it difficult to distinguish the differences. The authors should maximize contrast and use different colors only when it is necessary to understand the experiment.

Yes, except for Fig. 2, the colours of histogram bars carry no information. We have now converted the colour of most diagrams to the same shade of grey or to a neutral colour.

Reviewer #1 (Remarks to the Author):

The authors have diligently addressed all my points and those of the other reviewers. I believe that the paper is now acceptable.

Reviewer #2 (Remarks to the Author):

I'm very supportive of this paper. It would've been nice to see the effect of supercoiling on H-NS proven by experimentation but I don't think this should be a deal breaker.